# Digital Health Literacy of Adolescents and Its Association with Vaccination Literacy: The First Evidence from Lithuania

**DOI:** 10.3390/epidemiologia6040073

**Published:** 2025-11-03

**Authors:** Kristina Motiejunaite, Gerda Kuzmarskiene, Saulius Sukys

**Affiliations:** Department of Physical and Social Education, Lithuanian Sports University, Sporto g. 6, LT-44221 Kaunas, Lithuania; gerda.kuzmarskiene@stud.lsu.lt (G.K.); saulius.sukys@lsu.lt (S.S.)

**Keywords:** adolescents, digital health literacy, vaccination literacy

## Abstract

Background: Health literacy, including its digital and vaccination-specific components, is essential for informed health decision-making in adolescence—a developmental period when health attitudes and behaviors are shaped and may persist into adulthood. Although the importance of these competencies is increasingly recognized, little is known about the relationship between digital health literacy (DHL) and vaccination literacy (VL) among adolescents. The aim of this study was to investigate the associations between DHL and VL among Lithuanian adolescents, and to evaluate the psychometric properties of the Digital Health Literacy Questionnaire (HLS19-DIGI) and the Vaccination Literacy Questionnaire (HLS19-VAC) in this population. Methods: A cross-sectional survey was conducted with 9–12 grades students from Lithuanian gymnasiums using HLS19-DIGI and HLS19-VAC instruments. Analyses included confirmatory factor analyses for validity, McDonald’s omega for reliability, ANOVA and chi-square tests for group comparisons, and linear regression to evaluate DHL—VL associations, adjusting for gender, grade, and frequency of digital resource use. Results: A total of 792 students (42.0% male; mean age 16.4 years) completed the survey. The HLS19-DIGI (CFI = 0.945, TLI = 0.923, RMSEA = 0.081) and HLS19-VAC (CFI = 0.986, TLI = 0.959, RMSEA = 0.089) showed satisfactory structural validity, and both scales had good reliability (0.757 and 0.803). Mean DHL and VL scores were 78.28 (SD = 24.24) and 82.64 (SD = 27.22), respectively. Over half of the participants had excellent DHL (55.7%) and VL (63.4%). DHL was a strong predictor of VL (β = 0.429, *p* < 0.001). The frequency of digital resource use was not significantly related to VL. Conclusions: Higher DHL is associated with higher VL, suggesting that skills in searching for, appraising, and applying online health information can enhance informed vaccination decision-making. Interventions delivered through trusted channels, such as schools and healthcare providers, should aim to strengthen both literacies, address misinformation, and foster critical evaluation competencies to support vaccine uptake in youth.

## 1. Introduction

Health literacy represents a key determinant of personal and population health, reflecting individuals’ ability to obtain, interpret, evaluate, and apply health information and services to make informed decisions and maintain well-being [1,2,3]. Individuals with adequate health literacy tend to adopt preventive health behaviors, manage chronic conditions more effectively, adhere to medical advice, and communicate productively with healthcare professionals [2,4,5]. Furthermore, adequate health literacy has been linked to fewer hospital admissions and lower reliance on emergency care, alongside better self-rated health and enhanced overall quality of life [4,6]. In contrast, limited health literacy corresponds to increased risks of adverse health outcomes, including multimorbidity, lower self-rated health, less frequent use of healthcare services, and heightened susceptibility to misinformation [7,8]. Evidence consistently demonstrates that individuals with inadequate health literacy exhibit a heightened propensity for erroneous health-related judgments and participation in deleterious health behaviors—including tobacco consumption and insufficient physical activity—which collectively contribute to increased disease burden and premature mortality [4,9].

Digital health literacy (DHL) refers to the capacity to search for, access, comprehend, and evaluate health information from digital sources, as well as to apply this knowledge in addressing or resolving health-related issues [10,11,12]. It builds upon general health literacy by incorporating the digital competencies required to navigate online health information and digital healthcare services effectively. Adequate DHL enables individuals to locate and use online health resources, critically assess the credibility of digital content, and employ digital tools for disease prevention, self-management, and informed decision-making [13,14]. Evidence from a systematic review [13] indicates that interventions designed to improve DHL– such as e-learning programs, mobile health applications, and community-based digital training—enhance self-care practices, medication adherence, and clinical outcomes, while also increasing access to healthcare and reducing health disparities. Conversely, limited DHL is associated with greater susceptibility to misinformation, lower engagement with digital health interventions, and poorer health outcomes, particularly among individuals with chronic conditions or lower socioeconomic status [14].

The significance of health literacy, and particularly DHL, becomes most evident in adolescence—a pivotal stage of development when health attitudes and behaviors are formed and often carried into adulthood [15]. Adequate health literacy empowers adolescents to take responsibility for their own health, make informed decisions in daily life, and effectively navigate healthcare systems, thereby fostering the adoption of healthy behaviors and lifestyles [16,17,18].

Adolescents today are immersed in a digital environment and confronted with an overwhelming volume of online health information. DHL has therefore emerged as a crucial competency for this population. While many adolescents are highly confident in their digital skills, research consistently finds that their ability to evaluate the reliability and accuracy of online health information critically lags behind these technical skills [19]. However, the ability to effectively navigate, assess, and apply this information differs considerably within this population [18,20,21]. This gap has been documented across various studies, which show that although young people frequently use the internet and social media for health information, they often demonstrate limited capacity to critically appraise and utilize the information obtained [20,22]. Adolescents also tend to overestimate their DHL [20] and, despite expressing skepticism, continue to use online health resources [23]. Evidence from Germany indicates that a substantial proportion of adolescents have problematic or insufficient DHL, with competencies influenced by age and sociodemographic factors [19]. In Australia, although adolescents report relatively high levels of DHL, they often struggle with complex terminology and rarely act on online health information without consulting parents or professionals—underscoring the need to strengthen critical health literacy [20]. Similarly, in Norway, more than half of adolescents demonstrate low or insufficient DHL, with competencies influenced by age and sociodemographic factors [24].

Turning to the Lithuanian context, research indicates adolescent health literacy remains sub-optimal, with many lacking the necessary skills to access, understand, evaluate, and apply health information in daily life [25,26,27]. Nationwide studies employing validated instruments, such as the HLS_19_-Q12, report that although approximately two-thirds of Lithuanian adolescents demonstrate sufficient or excellent health literacy (67.1%), almost one-third exhibit problematic or inadequate levels (27.7% problematic; 5.2% inadequate) [25]. Additional investigations reveal that only about 17–22% of Lithuanian adolescents achieve an adequate level of health literacy, with the majority facing particular challenges in critically analyzing health information within digital environments [26,27].

In recent years, growing emphasis has been placed on the role of health literacy in supporting vaccination, since the capacity to obtain, interpret, evaluate, and use trustworthy health information is crucial for making well-informed immunization decisions [17]. Within this framework, vaccine literacy has emerged as a specific application of health literacy, particularly relevant in the context of widespread misinformation and rising vaccine hesitancy [28]. Evidence shows that higher health literacy is linked to greater acceptance of vaccines, improved ability to evaluate vaccine-related information, and reduced susceptibility to misinformation, with trust in the healthcare system further reinforcing these positive attitudes [29,30]. In contrast, low health literacy can contribute to negative attitudes toward vaccination, increased indecision, or outright refusal to vaccinate, which may ultimately have adverse consequences for public health [31]. As digital natives, adolescents frequently seek health information online and through school-based sources, including vaccine-related content [32,33]. Simultaneously, vaccination literacy (VL)—defined as the knowledge and skills required to make informed immunization decisions [28],—has gained particular importance in light of global public health challenges and the widespread circulation of vaccine-related misinformation. A systematic review of qualitative studies [34] showed that adolescents frequently demonstrated only a limited understanding of various vaccines and often believed that information about vaccines was for parents and not for themselves. Even when independent consent was possible, adolescents remained reliant on their parents for vaccination decisions [34].

Despite its growing relevance, VL remains suboptimal among adolescents. Evidence from systematic reviews indicates that broader knowledge of vaccine-preventable diseases, more substantial confidence in vaccines, and active participation in decision-making are linked to higher vaccination uptake, while low awareness and lack of sufficient information act as barriers to acceptance [35]. Adolescents most frequently rely on parents and healthcare professionals for vaccine information and generally place more trust in these sources than in social media [36,37,38]. Qualitative studies further emphasize that exposure to online misinformation and limited vaccine literacy promote hesitancy, while family, peer, and community norms substantially shape vaccination attitudes [39]. In addition, socio-economic circumstances, cultural beliefs, and mental health conditions—such as attention-deficit/hyperactivity disorder—also exert an influence on vaccine perceptions and confidence [35,39,40]. Collectively, these findings underscore the pressing necessity to strengthen digital and vaccine literacy, counter misinformation, foster trust in healthcare systems, and actively engage adolescents in vaccination decision-making.

In summary, adolescents face distinctive challenges in the digital environment: although they often demonstrate strong basic digital and vaccination literacy, they frequently lack critical appraisal skills. This discrepancy is compounded by reliance on parents and peers for vaccine-related decisions.

Although the importance of health literacy is increasingly recognized, the association between adolescents’ DHL and VL remains insufficiently examined. To address this gap, the present study aimed to investigate the associations between DHL and VL among Lithuanian adolescents, and to evaluate the psychometric properties of the Digital Health Literacy Questionnaire (HLS19-DIGI) and the Vaccination Literacy Questionnaire (HLS19-VAC) in this population. Based on previous evidence with university students [41], we hypothesized that DHL would be positively associated with VL. This hypothesis reflects the expectation that competencies in locating, evaluating, and applying online health information can directly improve adolescents’ ability to understand and make informed vaccination decisions.

## 2. Materials and Methods

### 2.1. Study Design, Setting, and Population

This was a cross-sectional design study conducted with the 9th–12th-grade students from Lithuanian gymnasiums. In the Lithuanian education system, a gymnasium refers to an upper secondary school offering general education for students in grades 9–12 (approximately ages 15–19). The inclusion criteria for participation in the study were as follows: enrollment in Lithuanian gymnasiums (grades 9–12) during the data collection period (January–March 2024); age within the typical range for gymnasium students (approximately 15–19 years); provision of written informed consent by parents or legal guardians, with subsequent individual assent from the adolescent participants; sufficient proficiency in Lithuanian to comprehend and complete the study questionnaire, which was administered exclusively in Lithuanian; and the capability to independently complete the online survey via the Google Forms platform. Exclusion criteria encompassed failure to obtain parental consent or individual assent, incomplete responses or responses containing technical errors, and insufficient proficiency in Lithuanian to complete the survey instrument.

The required sample size was calculated based on the national statistical data about the total population of students enrolled in grades 9–12 in Lithuanian gymnasiums during the 2023/2024 academic year. Assuming a 95% confidence level, a 5% margin of error, and an anticipated response distribution of 50%, the minimum sample size required was 384 students. According to the official data from the Ministry of Education, Science and Sport of Lithuania, a total of 102,341 students were enrolled in grades 9–12 in Lithuanian gymnasiums on 1 September 2023 [42]. The sample size was calculated for this finite population using a 95% confidence level, a 5% margin of error, and an anticipated response distribution of 50%. This proportion represents the most conservative estimate, commonly applied when there are no prior data on the expected variability of the study variable, as it maximizes the required sample size [43]. The final analytic sample comprised 792 adolescents, exceeding the minimum threshold and thereby ensuring adequate statistical power and population representativeness.

### 2.2. Ethical Considerations

Ethical approval for the study was secured from the University Ethics Committee. Information sheets accompanied by parental consent forms were then distributed to parents or legal guardians. Upon receipt of parental consent, students were also provided with their own consent forms. These documents outlined the study’s objectives and procedures, emphasized the voluntary nature of participation and the right to withdraw at any time, assured participants of anonymity, and included the contact information of the principal investigator.

### 2.3. Recruitment

The study participants were selected from Lithuanian gymnasiums using a multistage cluster sampling technique. First, gymnasiums were chosen from all Lithuanian regions. The research team then contacted the principals of the selected schools. If the school administration did not permit the survey to be conducted, another gymnasium from the same cluster was selected. From each chosen gymnasium, we randomly selected samples of 9th–12th-grade classes. After obtaining consent from both parents and students and agreeing on the date and time of the survey, the research team visited the school. Data collection took place between January and March 2024, during regular school hours, using an online Google Forms questionnaire. On average, students required approximately 20 min to complete the questionnaire.

### 2.4. Measures

#### 2.4.1. Digital Health Literacy

To assess DHL, the Digital Health Literacy Questionnaire (HLS_19_-DIGI) was utilized, which was originally developed by the Health Literacy Population Survey 2019–2021 [12]. The HLS_19_-DIGI instrument comprises an 8-item scale to assess DHL itself (HL-DIGI), a 2-item subscale evaluating interaction with digital devices (HL-DIGI-INT), and a 6-item subscale measuring the frequency of digital resource use (HL-DIGI-DD) [12,44]. In this study, we did not use HL-DIGI-INT. Originally, HLS_19_-DIGI was developed to measure adult populations, but it was recently found to be a valid instrument for measuring DHL in adolescents [24]. The 8-item HL-DIGI questionnaire was prefaced with the statement, “When you search online for information on health, how easy or difficult is it for you?” Participants responded on a four-point scale, from 1 “Very difficult” to 4 “Very easy”.

In the data analysis, response distributions for each item were first summarized by the percentage of participants selecting “very difficult” or “difficult.” Subsequently, we calculated the average frequency with which each respondent chose each response category for all items. The overall HL-DIGI score was then determined as the percentage (ranging from 0 to 100) of items answered with “very easy” or “easy”, provided that at least 80% of items contained valid responses [12,44]. Higher scores indicate a higher level of DHL. Finally, we calculated the DHL levels. Categorizing levels were below 50 (inadequate); between 50 and 66.66 (problematic); between 66.67 and 83.33 (sufficient); above 83.34 (excellent).

To assess the frequency of digital resource usage (HL-DIGI-DD), participants were asked: “In a typical week, on how many days do you use the following digital resources to obtain health-related information?”. The list of resources (websites, social media, digital devices, mobile health apps, eHealth, or other digital resources) was presented. Respondents could select from the following options: “Not relevant for me”, “Less than once per week”, “1–3 days per week”, “4–6 days per week”, “Once a day”, or “More than once per day”.

#### 2.4.2. Vaccination Literacy

VL was measured using the HLS_19_-VAC instrument, developed by the same M-POHL project group [12,45]. HLS_19_-VAC is a four-item scale measuring knowledge, motivation, and skills to find, understand, evaluate, and apply immunization-related information, aiming to make better immunization decisions. Each item was rated on a four-point scale, ranging from 1 “Very difficult” to 4 “Very easy.” The analytical procedures mirrored those used for DHL, including summarizing response distributions for each item, calculating the average frequency of each response category across all items for each participant, determining the overall VL score, and categorizing levels of VL.

### 2.5. Data Analysis

For all data analysis, we used IBM SPSS Statistics 29.0 (IBM, Armonk, NY, USA) and JASP 0.19.3 (University of Amsterdam, Amsterdam, The Netherlands). As the HLS_19_-DIGI and HLS_19_-VAC instruments had not been previously applied to an adolescent population in Lithuania, we first assessed their structural validity through exploratory factor analysis (EFA) and confirmatory factor analysis (CFA). To evaluate the suitability of the data for EFA, the Kaiser–Meyer–Olkin (KMO) test and Bartlett’s test of sphericity were conducted with a cut-off no less than 0.60 and a significant Bartlett’s test [46]. For CFA, given the ordinal nature of the 4-point Likert scale responses, the Diagonally Weighted Least Squares (DWLS) estimator was used. Model fit was evaluated using multiple indices: the root mean square error of approximation (RMSEA) of 0.08 or lower, the comparative fit index (CFI), and the Tucker–Lewis’s index (TLI) of 0.90 or higher [47,48]. We tested the one-factor solution based on the theoretical conceptualization of the HLS19 instruments. Alternative factor structures were not analyzed, as the single-factor model demonstrated adequate fit and aligned with previous validation studies [49]. Convergent validity of the scales was examined using Average Variance Extracted (AVE) and Composite Reliability (CR). A CR value of 0.70 or higher was regarded as adequate, whereas an AVE value higher than 0.5, together with a CR of ≥0.70, was considered acceptable [50]. Reliability of the HLS19-DIGI and HLS19-VAC scales was assessed using McDonald’s omega (ω). Descriptive statistics (mean, standard deviation, sample size/proportion, and correlations) were calculated. Variable normality was confirmed as skewness and kurtosis values remained within ±2 [51]. Analyses of variance (ANOVA) and chi-square (χ^2^) tests were conducted to compare DHL and VL across gender and grade. A two-way ANOVA (with Tukey post hoc comparisons) was applied to examine the interaction between gender and grade, and effect sizes were estimated using eta squared (η^2^) [52]. To further examine the relationship between DHL and VL, a multiple linear regression analysis was conducted. The dependent variable was the overall VL score (0–100). Independent variables included gender (coded 0 = female, 1 = male), grade (coded 9th–12th as ordinal variables), frequency of digital resource use (treated as an ordinal variable ranging from 1 = not relevant or less than once per week to 5 = more than once per day), and DHL (continuous score, 0–100). The selection of covariates was theoretically grounded in previous studies of adolescent health literacy and vaccine-related behaviors. Prior to running the regression, assumptions of linearity, independence, and homoscedasticity were checked. Multicollinearity was assessed using variance inflation factors (VIFs), with values < 5 indicating no issues. In our study, multicollinearity was an issue with the sociodemographic variables sibling and lives with (both parents, one parent, only grandparents, or foster home); therefore, these variables were excluded from the final model. Model fit was evaluated by the adjusted R^2^ and F-statistic. Effect sizes were reported as standardized beta coefficients (β), with statistical significance set at *p* < 0.05.

## 3. Results

### 3.1. Sociodemographic Characteristics

In this study, a total of 825 students participated, but 33 completed survey questionnaires with errors or with no answers to all questions. For the analysis, we used data from 792 adolescent students. Among study participants, 42.0% were male and 58% female with a mean age of 16.4 years (SD = 1.11) (Table 1).

### 3.2. Digital Health Literacy

#### 3.2.1. Validity and Reliability Analyses

Before presenting the results of adolescents’ DHL, construct validity and reliability of the HL-DIGI were examined. The one-dimensional structure of the HL-DIGI was evaluated using Confirmatory Factor Analysis (CFA). This analysis revealed an overall good-of-fit based on fit indices: RMSEA = 0.081 (95% CI = 0.071, 0.099), CFI = 0.945; TLI = 0.923.

The composite reliability of the scale was 0.897. Convergent validity was also supported, with an AVE value of 0.522 exceeding the recommended threshold. In addition, the reliability of the HLS19-DIGI was confirmed by McDonald’s omega (ω = 0.757), which demonstrated acceptable reliability.

#### 3.2.2. Descriptive, Gender, and Grade Differences in DHL Among Adolescents

The DHL package encompasses skills such as searching for, accessing, comprehending, evaluating, verifying, and utilizing online health information, assessed through eight specific items of the HL-DIGI scale. The proportion of combined ‘difficult’ and “very difficult” responses ranged from 9.1% for the easiest item (“visiting different websites to check whether they provide similar information on a topic”) to 35.8% for the most difficult item (“judging whether the information is reliable”) (Appendix A). On average, adolescents most frequently reported that HL-DIGI items were “easy” (53%) or “very easy” (25%). Fewer than one in five (19%) rated the items as “difficult”, and only 3% as “very difficult”.

DHL scores were calculated on a scale from 0 to 100, with a mean score of 78.28 (SD = 24.24). A two-way ANOVA was performed to examine gender and grade differences in DHL among adolescents. Gender and grade were independent variables, and the DHL score was a dependent variable. The results revealed a nonsignificant gender-grade interaction (F3, 792 = 1.379, *p* = 0.248 (Table 2). Study results found a significant gender effect, indicating gender differences by comparing DHL scores (F1, 792 = 7.312, *p* = 0.007, ηp2 = 0.01). Results also showed a significant grade effect on DHL scores (F3, 792 = 3.972, *p* = 0.008, ηp2 = 0.02). Turkey post hoc analyses revealed that 11th grade schoolchildren scored higher (M= 83.283, SD = 21.389), than 9th grade (M = 75.845, SD = 25.089) (*p* = 0.008) and 10th (M = 76.672, SD = 24.799) (*p* = 0.026) grade students.

We also assessed the levels of DHL among adolescents (Table 2). It was found that more than half (55.7%) of adolescents demonstrated a high level of DHL, while 14.5% had a sufficient level. The remaining adolescents were classified as having low levels (19.4% problematic and 10.4% inadequate). No statistically significant differences were observed in DHL levels when compared by gender (ꭓ^2^ (3, 792) = 7.19, *p* = 0.066) or by grade level (ꭓ^2^ (9, 792) = 13.17, *p* = 0.155).

We measured the frequency with which adolescents used different digital sources and resources for health promotion (Table 3). Social media and websites were reported as the most frequently used, whereas digital interaction with the health system was the least common. The mean score for overall use of digital resources was 2.09 (SD = 0.91). No statistically significant differences were observed in the use of digital sources and resources for health by gender and grade.

### 3.3. Vaccination Literacy

#### 3.3.1. Validity and Reliability Analyses

Before presenting the results of adolescents’ VL, construct validity and reliability of the HLS_19_-VAC were examined. The one-dimensional structure of the HLS_19_-VAC was evaluated using CFA. RMSEA = 0.089, CFI = 0.986; TLI = 0.959. Composite reliability was 0.809, and convergent validity was confirmed with an AVE of 0.501. In addition, the reliability of the HLS_19_-VAC was confirmed by McDonald’s omega (ω = 0.803), demonstrating acceptable reliability.

#### 3.3.2. Descriptive, Gender, and Age Differences in VL Among Adolescents

The VL package encompasses skills related to finding, understanding, judging, and applying vaccination information in order to make informed immunization decisions, assessed through four specific items. The proportion of combined “difficult” and “very difficult” responses ranged from 10.1% for the easiest item (“understanding why you or your family may need vaccinations”) to 20.3% for the most difficult item (“judging which vaccinations you or your family may need”) (Appendix A). On average, adolescents most frequently rated the items as “easy” (53%) or “very easy” (30%), while fewer reported “difficult” (15%) or “very difficult” (2%).

We also calculated VL scores ranging from 0 to 100, with a mean of 82.64 (SD = 27.22). A two-way ANOVA was conducted to examine gender and grade differences in VL among adolescents, with gender and grade as independent variables and the VL score as the dependent variable. The analysis revealed a nonsignificant gender-grade interaction (F3, 792 = 1.069, *p* = 0.362 (Table 4). However, a significant main effect of gender was found, with females scoring higher on VL than males (F(1, 792) = 7.709, *p* = 0.006, ηp^2^ = 0.011). Results also showed a significant grade effect on VL scores (F3, 792 = 5.136, *p* = 0.002, ηp2 = 0.022). Tukey post hoc comparisons showed that 11th-grade students scored significantly higher (M = 88.06, SD = 23.50) than 9th-grade students (M = 77.58, SD = 29.57; *p* < 0.001).

We also assessed the levels of VL among adolescents (Table 4). It was found that nearly two out of three adolescents (63.4%) demonstrated excellent, and 15.9% adequate levels of VL. The remaining adolescents exhibited problematic (12.2%) or inadequate (8.5%) literacy levels. Statistically significant differences were identified when comparing VL levels by gender (ꭓ^2^ (3, 792) = 8.57, *p* = 0.036) or by grade (ꭓ^2^ (9, 792) = 18.775, *p* = 0.027).

### 3.4. Association Between DHL and VL

First, a correlational analysis revealed a significant positive association between DHL and VL (r = 0.44, *p* < 0.001). To further examine this relationship, a multiple linear regression analysis was conducted with the overall VL score as the dependent variable. Independent variables included gender, grade, frequency of digital resource use, and DHL This combination of variables significantly predicted VL *F* (4, 689) = 44.533; *p* < 0.001) (Table 5) and explained 20.1% of the variance in VL (adjusted R^2^ = 0.201). Assumptions of linearity, independence, and homoscedasticity were met. Multicollinearity diagnostics confirmed that all variance inflation factors (VIFs) were close to 1, indicating no issues with collinearity. Study results showed that grade was a significant predictor of VL (β = 0.085, *p* = 0.013), whereas gender did not reach statistical significance (β = 0.058, *p* = 0.092). Frequency of digital resource use was also not associated with VL (β = 0.011, *p* = 0.998). We found that DHL was a significant predictor of VL (β = 0.429, *p* < 0.001), accounting for the largest share of explained variance.

## 4. Discussion

To our knowledge, this study is among the first to examine how DHL relates to VL in a large, school-based adolescent sample using the HLS_19_-DIGI and HLS_19_-VAC instruments. The analysis yielded three principal findings. First, adolescents reported generally high DHL and VL, yet meaningful proportions still experienced difficulty with core evaluative tasks (e.g., judging reliability). Second, DHL emerged as a strong predictor of VL, whereas the frequency of digital resource use was not associated with VL. Third, grade level, but not gender, showed small, independent associations with VL. Together, these results suggest that how adolescents process online health information matters more than how often they are online. Additionally, but no less importantly, the study revealed good psychometric characteristics of the HLS19-DIGI and HLS19-VAC instruments. Specifically, our study contributes to the tiny number of studies that have confirmed that the HLS19-DIGI is a valid instrument for measuring DHL in adolescents [24]. In Lithuania, the latter questionnaire has already been adapted for adults [53], and our study showed that it is also suitable for the adolescent population. The psychometric properties of the HLS19-VAC have been tested more extensively only in the adult population [12]. Our study suggests that it can also be used to assess VL in older adolescents.

The positive association between DHL and VL observed in our study aligns with research demonstrating that digital and eHealth competencies support vaccine-related decision-making. Evidence from young adults in a lower-middle-income setting indicates that higher eHealth literacy correlates positively with intentions to receive vaccines, even after statistical adjustment. However, vaccine literacy itself did not reach significance in that analysis [54]. Recent evidence beyond COVID-19 similarly indicates that higher vaccine literacy—particularly competence and decision-making dimensions—relates to lower hesitancy and greater influenza vaccination among youth and adults [55]. Our finding of no significant association for “digital use frequency” aligns with this pattern: exposure alone is insufficient; the competencies captured by DHL (search, evaluation, verification, application) appear to be the active ingredients. This distinction highlights that not all online engagement is equal. Passive consumption of health-related content—for instance, scrolling through social media feeds or encountering incidental health messages—may increase exposure but rarely strengthens understanding or decision-making. In contrast, adolescents with higher DHL engage more actively and critically: they verify sources, compare information across platforms, and reflect on its credibility before applying it to their own health decisions. Such active processing likely accounts for the stronger association between DHL and VL observed in our study, suggesting that fostering evaluative and reflective skills may be more impactful than simply increasing access to digital information. This finding underscores that interventions aiming to improve adolescents’ vaccination literacy should not focus merely on increasing their exposure to digital health information, but rather on strengthening their competencies to critically appraise, interpret, and apply it. In practice, this means that educational programs in schools and community settings should move beyond teaching basic digital navigation skills and instead prioritize training in critical thinking, source verification, and evaluation of reliability. Such skill-oriented approaches may be more effective in reducing susceptibility to misinformation and fostering informed vaccination decisions than simply enhancing access to digital resources.

When placed in the context of international research, our results indicate that Lithuanian adolescents scored substantially higher on DHL than peers in several other countries. For instance, a recent Norwegian survey utilizing the HLS_19_-DIGI instrument found that over half of adolescents and young adults demonstrated limited DHL, with 54% classified at or below the basic proficiency level [24]. Similarly, a 13-country European study of adults revealed that between 22% and 58% of respondents considered it “difficult” or “very difficult” to locate, understand, and evaluate online health information [49]. These cross-national differences may stem from several factors. Variation in measurement instruments, thresholds, and scoring criteria can influence classification rates. Additionally, our data collection occurred in the post-pandemic period, when digital engagement in Lithuania—especially within educational settings—may have reinforced online information skills among adolescents. Differences in national school health education curricula and extracurricular health promotion activities could further contribute to elevated competence levels. Moreover, Lithuania’s highly connected youth culture and high internet penetration rates suggest adolescents may have more frequent exposure to diverse online health information environments [56,57]. While these contextual factors plausibly explain observed differences, it remains essential to address persistent gaps in critical evaluation skills, as reflected by the proportion of participants who reported difficulties in judging the reliability of online health content.

The significant positive associations between DHL and VL observed in our study indicate that adolescents adept at finding and evaluating health information online also tend to possess better knowledge and understanding regarding vaccines. Such an association is not unexpected—different facets of health literacy often co-occur. Prior research has documented that eHealth literacy is closely related to general health literacy in adolescents [58]. Our findings extend this pattern to the domain of vaccination. Navigating vaccination information likely draws on the same core skills involved in digital literacy –such as critical reading, source verification, and applying health knowledge. It follows that enhancing one literacy domain may positively impact the other. Evidence from international surveys suggests that individuals with higher vaccination-specific literacy tend to have more positive attitudes toward vaccines [35]. Thus, the association observed in our study reinforces the idea that building digital literacy could be a strategic lever to improve VL, ultimately supporting informed vaccination decisions among youth.

Our analysis did not find significant gender differences in DHL level. This result concurs with several international observations, as gender was not a consistent predictor of DHL disparities in the 13-country European survey [49]. Equal access to technology and health education in schools might be contributing to a leveling of the playing field. Our findings imply that interventions to further improve DHL or VL need not be strongly gender-targeted in this age group—both male and female adolescents are equally in need of literacy strengthening initiatives.

Placing the current findings within the broader body of VL research reveals both encouraging and cautionary perspectives. The relatively high levels of VL observed among Lithuanian adolescents are a positive sign, particularly in light of evidence identifying limited VL as a barrier to vaccine uptake. A recent systematic review of studies on COVID-19 vaccination in adolescents found that insufficient vaccine literacy is associated with less favorable attitudes toward vaccination [39]. In contrast, enhancing adolescents’ knowledge and competencies related to vaccination has the potential to foster more informed and confident decision-making. According to Moreira da Cunha et al. [39], adolescent vaccine acceptance can be strengthened through targeted, accessible literacy interventions, complemented by initiatives aimed at building trust and supporting autonomy.

However, these promising outcomes must be balanced with an awareness of persistent challenges. Notwithstanding the generally high mean scores, our findings should be interpreted with some caution. “Relatively high” literacy averages do not guarantee that all individuals are fully competent in every aspect of DHL or VL. Other research underscores specific challenges even digitally literate youth face. Notably, the most demanding tasks in DHL involve critical evaluation. For example, judging whether online health information is reliable or influenced by commercial interests tends to be the most complex challenge across populations [49]. Adolescents may be proficient in using search engines and social media, yet still face challenges in distinguishing credible information from misinformation. A Norwegian study found that adolescents often rely on family and friends as information sources and may not consistently practice verifying the credibility of online content [24]. Thus, even with relatively high average scores among Lithuanian adolescents, fostering strong critical thinking and source evaluation skills should remain a central goal of literacy development.

By comparing our cohort with international benchmarks, it becomes clear that continuous investment in DHL and VL is warranted. Such efforts will help adolescents in Lithuania and elsewhere to confidently navigate digital health resources and make well-informed decisions about vaccination, ultimately supporting better health outcomes at both individual and community levels. Integrating these competencies into national school health education and youth health promotion policies could represent a sustainable strategy to strengthen adolescent health literacy overall and to ensure long-term public health resilience.

### Limitations and Future Directions

Several limitations should be considered when interpreting the findings of this study. First, the cross-sectional design precludes causal inference regarding the directionality of the relationship between DHL and VL. Second, the study sample, while sizeable, was geographically restricted to Lithuania, which may limit the generalizability of the findings to adolescents in other cultural and socio-economic settings. Third, the regression model included a restricted set of covariates. Additional demographic data, such as parental education, socioeconomic background, or detailed patterns of internet use, were not available in our dataset. We recognize this as a limitation and recommend that future studies incorporate a broader set of variables to build more robust explanatory models. This is also important when considering the social desirability factor, particularly when studying a sensitive topic such as vaccination, as social norms and family or peer pressure may influence adolescents. Fourth, unmeasured confounding variables, such as prior exposure to health education programs or parental influence, may have affected the observed associations. Both DHL and VL were self-reported, capturing perceived rather than actual competencies, which may introduce reporting bias. Future research employing longitudinal designs and objective performance-based measures could provide a more comprehensive understanding of these relationships. Finally, we examined the structural validity of the CFA-applied measures for DHL and VL. Although most model fit indices were good, the RMSEA value for the HLS19-VAC scale was slightly above the commonly accepted. This slightly elevated RMSEA may reflect age-related cognitive and experiential factors specific to adolescents, such as limited exposure to vaccination decision-making or varying understanding of immunization terminology. These factors could lead to greater response variability and marginally lower model fit compared with adult samples, but do not substantially compromise the structural validity of the instrument. Therefore, in future studies using this research instrument, we would recommend additionally checking all model fit indices, with particular attention to RMSEA. Also consider using different estimators.

## 5. Conclusions

This study reveals a positive link between digital and vaccination literacy in adolescents, suggesting that stronger skills in finding, evaluating, and applying online health information support informed vaccination decisions. Targeted interventions—delivered via trusted sources—should build both literacies, combat misinformation, and strengthen critical evaluation skills to improve vaccine uptake among youth.

## Figures and Tables

**Table 1 epidemiologia-06-00073-t001:** Sample characteristics (*n* = 792).

Variable	Categories	*n*	%/M (SD)
Age		792	16.4 (1.11)
Sex	Male	333	42.0
	Female	459	58.0
Grades	9th grade	225	28.4
	10th grade	217	27.4
	11th grade	202	25.5
	12th grade	148	18.7
Sibling	0	272	34.3
	1	382	48.2
	2	92	11.6
	3	22	2.8
	4	17	2.1
	<4	7	0.9
Lives with	With both parents	634	81.1
	With one parent	139	17.8
	Only with grandparents	6	0.8
	At the foster home	3	0.4

**Table 2 epidemiologia-06-00073-t002:** Descriptive statistics for DHL across gender and grades.

Gender	Grades	DHL Mean (SD)	Levels of DHL, %			
			Inadequate	Problematic	Adequate	Excellent
Male (*n* = 333)	9th	79.61 (22.17)	7.4	22.1	15.8	54.7
	10th	76.01 (26.28)	12.4	19.1	15.7	52.8
	11th	86.39 (18.97)	3.3	13.0	12.0	71.7
	12th	82.38 (21.60)	7.0	15.8	17.5	59.6
	Total	80.99 (22.70)	7.5	17.7	15.0	59.8
Female (*n* = 459)	9th	73.08 (26.78)	16.2	20.8	14.6	48.5
	10th	77.14 (23.81)	11.7	21.1	14.8	52.3
	11th	80.69 (22.98)	7.3	20.0	12.7	60.0
	12th	74.46 (26.57)	14.3	20.9	14.3	50.5
	Total	76.31 (25.20)	12.4	20.7	14.2	52.7

Note: %: percentage; SD: standard deviation; DHL: digital health literacy.

**Table 3 epidemiologia-06-00073-t003:** Distribution of responses on the use of digital health resources.

Type of Digital Resources	Not Relevant or Less than Once per Week	1–3 Days per Week	4–6 Days per Week	Once a Day	More than Once per Day
Websites	33.8	23.0	12.4	7.4	23.4
Social media, including online forums	36.6	19.6	12.5	7.3	24.0
A digital device related to health or healthcare	45.6	19.2	10.6	10.2	14.4
Health app on your mobile phone	51.0	19.2	10.1	8.8	10.9
Digital interaction with your health system	73.1	13.3	5.6	3.9	4.1
Other	82.8	8.6	2.8	1.8	4.0

Note: distribution in percentages.

**Table 4 epidemiologia-06-00073-t004:** Descriptive statistics for VL across gender and grades.

Gender	Grades	VL Mean (SD)	Levels of VL, %			
			Inadequate	Problematic	Adequate	Excellent
Male (*n* = 333)	9th	80.12 (28.36)	9.6	15.7	15.7	59.0
	10th	83.90 (26.47)	5.5	15.1	13.7	65.8
	11th	89.20 (20.52)	2.5	9.9	14.8	72.8
	12th	92.71 (13.60)	0	4.2	20.8	75.0
	Total	85.79 (24.10)	4.9	11.9	15.8	67.4
Female (*n* = 459)	9th	75.68 (30.42)	12.6	18.0	18.0	51.4
	10th	79.87 (28.54)	11.0	13.6	16.9	58.5
	11th	87.11 (25.80)	7.2	6.2	13.4	73.2
	12th	79.82 (30.36)	13.3	10.8	14.5	61.4
	Total	80.44 (29.02)	11.0	12.5	15.9	60.6

Note: %: percentage; SD: standard deviation; VL: vaccination literacy.

**Table 5 epidemiologia-06-00073-t005:** Linear Regression analysis on the VL.

Independent Variables	β	*p*	95% CI		VIF	AdjR^2^	F	*p*
			Lower	Upper				
						0.201	44.533	<0.001
Gender	0.058	0.092	−0.518	6.892	1.010			
Grades	0.085	0.013	0.462	3.835	1.008			
Use of digital resources	0.011	0.998	−1.941	1.968	1.014			
DHL	0.429	<0.001	0.407	0.560	1.031			

Note. Data presented in standardized β. Independent variables: gender (0 = female; 1 = male), grade (1 = 9th; 2 = 10th; 3 = 11th; 4 = 12th), use of digital resources from 1 = not relevant or less than once per week to 5 = more than once per day. Dependent variable: VL score: 0 = minimal to 100 = maximal.

## Data Availability

The data that support the findings of this study are available from the corresponding author, [K.M.], upon reasonable request.

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
