# Peer review of "Digital Health Literacy of Adolescents and Its Association with Vaccination Literacy: The First Evidence from Lithuania"

_epidemiologia, 2025, doi:10.3390/epidemiologia6040073_

Round 1
Reviewer 1 Report
Comments and Suggestions for Authors
Please see my uploaded review document.

Reviewer 2 Report
Comments and Suggestions for Authors
The work is interesting and focuses on digital health literacy and its use for vaccine literacy in the adolescent population. The methodology is correct, and I believe the modeling procedures have adequate theoretical support that supports the theoretical model analyzed. However, it would be advisable to add whether more models were analyzed and the reasons for not selecting them, or if only this single theoretical model was used.
However, caution should be exercised in not attributing a cause-and-effect relationship. The study is based on a cross-sectional design. Although the results identify a significant statistical association or correlation between variables, causality cannot be demonstrated.
Most studies related to digital literacy and responsible use of digital technology have this weakness. I believe it is important to encourage researchers to conduct follow-up, longitudinal, or experimental studies that can help confirm these types of results.
It would also be advisable to include social desirability bias among the study's limitations. This occurs when participants in a survey or self-report study answer questions in a way that makes them appear more competent, knowledgeable, or socially acceptable than they actually are. In this particular study, two factors could contribute to this bias: the self-report method used and a sensitive topic such as vaccine literacy, as adolescents may be influenced by social norms and family or peer pressure. They may feel the need to respond that they are more competent or knowledgeable than they actually are to conform to what they perceive as the acceptable response on the topic of vaccines.
Reviewer 3 Report
Comments and Suggestions for Authors
This manuscript presents a timely and relevant cross-sectional study examining the association between digital health literacy (DHL) and vaccination literacy (VL) in a large sample of Lithuanian adolescents. The study addresses an important gap in the literature by using validated instruments (HLS19-DIGI and HLS19-VAC) and rigorous statistical analysis.
The authors find that while the mean levels of DHL and VL are relatively high, a significant portion of adolescents still face difficulties, particularly with critically evaluating information. The key finding is that DHL is a strong predictor of VL, whereas the mere frequency of digital resource use is not. This is an important conclusion with clear implications for public health and health education.
The study is well-structured, the writing is clear, and the discussion effectively contextualizes the findings within the international literature. However, there are a few critical points that require clarification and significant revision before the manuscript can be considered for publication.
Major Comments
-
The most critical issue concerns the variance explained in the regression analysis. Both the Abstract and the Results section state that DHL "explain[s] 43% of variance" in VL. However, Table 5 reports an Adjusted R² for the entire model of 0.201, which corresponds to 20.1% of the variance. This value represents the variance explained by all variables in the model (gender, grade, digital resource use, and DHL), not just DHL alone. The 43% figure appears to be a misinterpretation of the standardized beta coefficient (β = 0.429). Please correct this discrepancy throughout the manuscript and base the interpretation of the association's strength on the correct metric (likely the adjusted R² for the full model or the change in R² attributable to DHL). This correction is fundamental, as the current statement significantly overstates the effect.
2. The methods section mentions the use of a "cluster sampling approach" but provides no details on how the schools (the "clusters") were selected. Were they chosen randomly from a list of all Lithuanian gymnasiums? Was it a convenience sample? Were they stratified by geographic region or socioeconomic status? This information is essential for assessing the representativeness of the sample and the generalizability of the findings, even within the Lithuanian context. Please add a more detailed description of the sampling strategy.
Minor Comments
1. In the validity analysis for the HLS19-VAC scale, the reported RMSEA is 0.089, which is slightly above the commonly accepted threshold of 0.08 for a good fit. While other indices (CFI, TLI) are strong, the authors might consider briefly acknowledging this slight weakness in the text, if only to demonstrate awareness of it.
2. The regression analysis found that the frequency of "Use of digital resources" was not a significant predictor of VL. This is a very interesting finding that strongly supports the paper's core message: that skill (DHL) is more important than mere exposure. In the Discussion section, this point could be further emphasized and elaborated upon, as it powerfully reinforces the need for educational interventions focused on critical appraisal skills rather than simply on access to information.
3. On page 8, in section 3.2.2, the text states: "We also assessed the levels of VL among adolescents (Table 3)". However, the data on VL levels (inadequate, problematic, etc.) are presented in Table 4. Please correct this reference
Round 2
Reviewer 1 Report
Comments and Suggestions for Authors
The authors have responded to my comments, however, there are still issues that need to be addressed.
- The explanation of the sample size calculation is vague. Please clarify the size of the target population and explain what does 50% response distribution mean? What reference did you use to anticipate 50%?
- The linear regression model is insufficient because it does not provide essential 95% confidence intervals. The column with t values can be omitted instead.
Reviewer 3 Report
Comments and Suggestions for Authors
While the manuscript is very strong, the following minor points might be considered to further enhance its quality:
Limitations Section: The authors rightly acknowledge that the RMSEA for the HLS19-VAC scale was slightly elevated (0.089). This transparency is commendable. While they recommend checking this in future studies, they could briefly add a sentence speculating on why this might have occurred in an adolescent sample or what the potential implications are for the model's fit.
Discussion: The discussion insightfully notes that the competencies captured by DHL appear to be the "active ingredients" for higher VL, rather than mere exposure to digital resources. This point could be slightly expanded by discussing the potential difference between passive consumption of health information online (e.g., through social media feeds) versus the active, critical engagement and evaluation skills that define DHL.
Self-Reported Measures: The limitations section is thorough, covering the cross-sectional design and the restricted set of covariates. The authors might also consider briefly mentioning the reliance on self-reported literacy as a potential limitation. Perceived skills can sometimes differ from objectively measured abilities, which is a common consideration in health literacy research
Minor Proofreading: There are very minor typographical errors in the tables that should be corrected. For example, in Table 4, the "Inadequate" percentage for 10th-grade males is written as "5.5.".
